# Rosiglitazone alleviates LPS-induced endometritis via suppression of TLR4-mediated NF-κB activation

**Hongchu Bao[1,2], Jianxiang Cong[1,2], Qinglan Qu[1,2], Shunzhi He[1,2], Dongmei Zhao[1,2], Huishan Zhao[1,2], Shuyuan Yin[1,2], Ding Ma**  **[1,2]***

**1** Reproductive Medicine Centre, Yantai Yuhuangding Hospital, Qingdao University, Yantai, China,
**2** Shandong Provincial Key Medical and Health Laboratory of Reproductive Health and Genetics (Yantai Yuhuangding Hospital), Yantai, China

* md0110@126.com

## Abstract

### Objective

The aim of this study was to investigate the anti-inflammatory effect of Rosiglitazone (RGZ) on lipopolysaccharide (LPS) -induced Endometritis and explore its possible mechanism.

### Methods

The preventive and therapeutic effects of RGZ on Endometritis were studied in vivo and in vitro. A total of 40 female C57BL/6 mice were randomly divided into the following 4 groups: RGZ+LPS, RGZ control, LPS and DMSO control. The mice uterine tissue sections were performed with HE and immunohistochemical staining. Human endometrial stromal cells (HESCs) were cultured, and different concentrations of LPS stimulation groups and RGZ and/or a TLR4 signaling inhibitor TAK-242 pretreatment +LPS groups were established to further elucidate the underlying mechanisms of this protective effect of RGZ.

### Results

The HE results in mice showed that RGZ+LPS group had less tissue loss than LPS group. Immunohistochemical staining (IHC) results showed that the expression of TLR4 after RGZ treatment was significantly lower than that in LPS group. These findings suggested that RGZ effectively improves the pathological changes associated with LPS-induced endometritis by inhibiting TLR4. Reverse transcription-polymerase chain reaction and western blot analysis demonstrated that RGZ pretreatment suppresses the expression of Toll-like receptor 4 (TLR4) and its downstream activation of nuclear factor-κB (NF-κB). In vitro, RGZ inhibited LPS-stimulated expression of proinflammatory cytokines in a dose-dependent manner and also downregulated LPS induced toll-like receptor 4 (TLR4) expression and inhibited phosphorylation of LPS-induced nuclear transcription factor-kappa B (NF-κB) P65 protein.

**Data Availability Statement:** All relevant data are within the paper and its Supporting Information files.

**Funding:** The study was supported by Shandong Medical and Health Science and Technology Development Project (No.202005030944). The study was supported by Shandong Medical and Health Science and Technology Development Project (No.202005030944). The funders had no role in study design, data collection and analysis, decision to publish, or preparation of the manuscript.

**Competing interests:** The authors have declared that no competing interests exist.

## Conclusions

These results suggest that RGZ may inhibit LPS-induced endometritis through the TLR4-mediated NF-κB pathway.

## Introduction

Endometritis not only affects embryo implantation to cause infertility but also leads to pregnancy-related diseases, such as miscarriage, premature rupture of membranes, premature birth, and intrauterine infection. Bacterial infection and dysbacteriosis are the causes of endometritis. Although the accepted method is antibiotic treatment, it has been reported that up to 24.6% of patients still have signs of endometritis after antibiotic treatment [1, 2]. Studies have shown that 91.4% and 70.7% of *E. coli* isolates isolated from cows with endometritis displayed resistance to cefotaxime and doxycycline, respectively [3].

The findings clearly demonstrate the importance of addressing antibiotic resistance. Therefore, it is of great significance to explore new drugs for the treatment of endometritis [4, 5]. As the main component of the outer membrane of Gram-negative bacteria, lipopolysaccharide (LPS) is a key factor in the induction of endometritis [6, 7]. LPS triggers the release of inflammatory factors, including interleukin-6 (IL-6), interleukin-1β (IL-1β) and tumor necrosis factor -alpha (TNF-α), to exacerbate endometrial tissue damage [8] and inhibit the production of cytokines to achieve therapeutic effects.

Toll-like receptors (TLRs) are a class of intracellular pattern recognition receptors (PRRs) that participate in the body's innate and adaptive immune responses by recognizing molecules such as LPS and membrane lipoproteins. TLR4 is an important member of the TLR family and can recognize the exogenous ligand LPS. Upon binding with LPS, TLR4 triggers intracellular signal transduction response, activating the NF-κB signaling pathway and ultimately leading to the generation of inflammatory mediators, which in turn elicit an inflammatory response [9, 10].

Rosiglitazone (RGZ) is a thiazolidinedione insulin sensitizer that is mainly used for treating type 2 diabetes. Studies have shown that RGZ not only sensitizes insulin but also possesses anti-inflammatory and immune-modulating properties. RGZ plays its anti-inflammatory roles in the acute and chronic inflammatory responses of many organs such as the heart, lung, kidney, and gastrointestinal tract [11–14]. Bo et al. reported that RGZ pretreatment alleviated LPS-induced placental inflammation and reduced embryonic mortality in mice [15]. Given that RGZ is known to exhibit anti-inflammatory effects, it is meaningful to study its role and potential mechanism in LPS induced endometritis. Therefore, this study aimed to establish an LPS-induced mouse endometritis model to determine the pathological changes in the mouse endometrium. Human endometrial stromal cells (HESCs) were stimulated with LPS and the anti-inflammatory mechanism of RGZ was assessed by pre-treatment with RGZ to detect TLR4, p-p65, p-IκBα, IL-1β and IL-6. The results of this study provide new ideas and avenues for the treatment of endometritis.

## Materials and methods

### Reagents

Rosiglitazone (BRL-49653) and a TLR4 signaling inhibitor TAK-242 (CLI-095) were purchased from MedChemExpress (NJ, USA) and dissolved in dimethyl sulfoxide (DMSO). LPS was purchased from servicebio (Wuhan, China). The NF-κB (p65) (F-6) antibody was

purchased from Santa Cruz (TX, USA). IκBα (L35A5) mouse mAb, p-IκBα (Ser32) (14D4) rabbit mAb and p-p65 (Ser536) (93H1) rabbit mAb were purchased from Cell Signaling Technology (Beverly, MA, USA). TLR4 (D220102) and GAPDH (D110016) antibodies were obtained from Sangon Biotech (Shanghai, China).

## Ethics statement

C57BL/6 mice [permit number: SCXK(LU)20190003)] were purchased from Jinan Pengyue Experimental Animal Breeding Co., Ltd. The animal experiment operation (approval number :2021S31 and 2022–255) complied with the requirements of the Experimental Animal Ethics and Welfare Committee of the Experimental Animal Center of Yantai Yuhuangding Hospital. All animal studies were conducted in strict accordance with the recommendations in the National Institutes of Health's Guidelines for the Care and Use of Laboratory Animals. All procedures were performed under anesthesia, and all mice were sacrificed through cervical dislocation, strictly following the three "Rs" principle to minimize the number and pain of experimental mice.

## Animals and treatment

C57BL/6 mice (40 females aged 8 weeks) were randomly divided into four groups (n = 10). The experimental protocol was based on a previous report [16]. 50uL LPS (1μg/mL) was intra-uterus infused induced endometritis as the paper described [17]. RGZ + LPS group: mice were intra-uterus infused with LPS two hours after intraperitoneal injection of RGZ (10 mg/kg) [18]. RGZ control group: mice were intra-uterus infused with phosphate buffered saline (PBS) two hours after intraperitoneal injection of RGZ. LPS group: mice were intra-uterus infused with LPS two hours after intraperitoneal injection of 200 μL PBS. DMSO control group received equal volumes of normal PBS. (RGZ solvent is DMSO, PBS containing 0.1% DMSO for intraperitoneal injection). After 24 hours, the mice were sacrificed by cervical dislocation, and the mouse uterus was collected for fixation, embedding, Hematoxylin and eosin (HE) staining and immunohistochemical staining.

## Cytokine assays

The uterus tissues of mice were homogenized with cold PBS at a ratio of 1:9 (w/v) and were centrifuged at 5000g for 5 min. The levels of IL-1β and IL-6 were detected in the supernatant, according to the manufacturer's instructions of ELISA kits.

## Histopathologic evaluation of the uterus tissue

All endometritis tissues isolated from mice subjected to cervical dislocation were fixed in 10% formalin, dehydrated in ethanol, and embedded in paraffin. After that, the tissue was sliced into 5μm sections and stained with HE. The sections were observed under an optical microscope (Leica, Wetzlar, Germany). Histology scoring was carried out according to the degree of injury such as the integrity of tissue structure and the number of infiltrated inflammatory cells.

## Immunohistochemistry

Immunostaining for TLR4 was performed on 5 μm sections of formalin-fixed, paraffin-embedded endometrioid biopsies. The slides were deparaffinized using fresh xylene and then rehydrated with gradient-grade ethanol. Antigen retrieval was carried out in a microwave oven at 100°C for 20 min, followed by cooling to ambient temperature. Next, the slides were incubated with 3% $H_2O_2$ for 10 min and blocked with 3% bovine serum albumin (BSA). The

sections were then incubated overnight at 4˚C with anti-TLR4 antibody at a dilution of 1:100. Afterward, they were incubated with horseradish peroxidase (HRP) conjugated goat anti-rat secondary antibody at a dilution 1:400 and at 37˚C for 30 min to amplify the signal. The sections were incubated with immunoreactivity complexes detected by 3, 3'-diaminobenzidine tetrahydrocholoride. The slides were dehydrated using a series of gradient-grade ethanol and xylene according to routine dehydration steps and finally fixed with neutral resin. Immunohistochemical images were captured using an optical microscope. The immunohistochemical staining was analyzed using ImageJ software to obtain the mean optical density (integrated optical density/total area). Four slides per mouse sample were randomly analysed for immunohistochemical quantification.

## Cell culture and treatment

Human endometrial stromal cells (HESCs, ATCC, Manassas, VA, USA) were cultured with phenol red-free DMEM/F12 (GIBCO, Grand Island, NY, USA) supplemented with 10% charcoal stripped fetal bovine serum (FBS) and 1% Penicillin-Streptomycin and maintained at 37˚C with 5% $CO_2$. Once HESCs reached 90% confluence, they were digested with 0.5% Trypsin and passed down from 1 to 2 or 3. The cultured HESCs were stimulated with LPS in a dose range of (0.01,0.05,0.25,1,5μg/mL) for 24 h to detect the secreation of inflammatory factors IL-1β and IL-6. To study the anti-inflammatory effects of RGZ, the cells were pretreated with various concentrations of RGZ (10 and 20 μM/L) for 30 min and then stimulated with LPS (1μg/mL) for 24 h. To clarify the decrease of the phospho-p65 (p-p65) NF-κB expression level and TLR4 signaling blockade, HESCs were obtained after 30min incubation with 10 μM RGZ alone or combined with 2 h incubation of 1 μM TAK-242 pretreatment and the followed by LPS induced for 24 h stimulation.

## CCK8 assay

Cell viability was determined by a Cell Counting Kit-8 (CCK-8) assay kit (sparkjade, China). HESCs were seeded at $0.3 \times 10^5$ cells/well in 96-well plates overnight and then treated with various concentrations of RGZ (0,5,10,20,40 μM/L) for 24 h. Subsequently, the cells were treated with a mixture of 100 μL of DMEM/F12 medium and 10% CCK-8 reagent. After 30min culture at 37˚C, the optical density (OD) of HESCs was immediately analyzed at 450 nm using a microplate reader (Thermo scientific, Waltham, MA, USA).

## RT-PCR

Total RNA was extracted from the obtained HESCs by TRIzol reagent according to routine procedures. Total RNA and complementary DNA were prepared in accordance with the manufacturer's instructions and analyzed on a real time PCR machine (FTC-3000P Funglyn Biotech, Toronto,Canada). The relative expression of each target gene was determined by comparing it with the corresponding β-actin threshold cycle (CT) values which were calculated using the $2^{-\Delta\Delta Ct}$ method. The primers of IL-1β, IL-6, TLR4 and β-actin were presented in Table 1.

## Western blot

Total protein was harvested from uterus tissues and cells using RIPA lysis buffer supplemented with a protease inhibitor. The total protein concentration was measured with a BCA protein assay kit (sparkjade, China). Next, equal amounts of protein were fractionated using 10% SDS-PAGE and then transferred onto polyvinylidene difluoride (PVDF, Millipore, USA) membranes. The membranes were blocked with 5% skim milk for 1 h and then incubated with the

**Table 1. Primers of human IL-1β, IL-6, TLR4 and β-actin.**

| Gene | Primer | Length (bp) |
|---|---|---|
| IL-1β (forward) | 5′– CAGAAGTACCTGAGCTCGCC –3′ | 153 |
| IL-1β (reverse) | 5′– AGATTCGTAGCTGGATGCCG –3′ | |
| L-6 (forward) | 5′–CCTCCAGAACAGATTTGAGAGTAG–3′ | 206 |
| IL-6 (reverse) | 5′– TGCGCAGAATGAGATGAGTT –3′ | |
| TLR4 (reverse) | 5′– GTCCTCAGTGTGCTTGTAGTATC –3′ | 160 |
| TLR4 (forward) | 5′– CATTCCTTACCCAGTCCTCATC –3′ | |
| β-actin (forward) | 5′– CTGGACTTCGAGCAAGAGATG –3′ | 182 |
| β-actin (reverse) | 5′– GAGTTGAAGGTAGTTTCGTGGA –3′ | |

indicated primary antibodies (1:1000 dilution) at 4˚C overnight. After incubation with horseradish peroxidase-conjugated second antibodies, the intensities of the proteins were analyzed with a Vilber Fusion FX5 Spectra (Vilber Lourmat, France). GAPDH served as an internal standard.

### Immunofluorescence

To study the anti-inflammatory effects of RGZ, the cells were pretreated with various concentrations of RGZ (10 and 20 μM/L) for 30 min thereafter stimulated with LPS (1μg/mL) for 24 h. HESCs were fixed with paraformaldehyde, permeabilized with PBS containing 0.3% Triton X-100, exposed to 3% BSA, and incubated with indicated antibodies at a 1:200 dilution at 4˚C overnight. Subsequently, the cells were incubated with FITC-labelled secondary antibodies at a 1:100 dilution for 1 h at 37˚C, and the nuclei were stained with DAPI dye for 10 min and visualized under a fluorescence microscope.

### Statistical analysis

All data were analyzed using GraphPad Prism 6. Statistical significance was determined using Student's *t*-test for two groups or one-way analysis of variance (ANOVA) for multiple group comparisons. The data were expressed as the means ± SEM. Values of $p < 0.05$ were considered statistically significant.

## Results

### Effect of RGZ on inflammatory cytokines

Inflammatory damage of the animals mainly occurs through releasing inflammatory cytokines. To assess the effects of RGZ on the levels of inflammatory cytokines induced by LPS in the uterine homogenates, the expression of IL-1β and IL-6 was detected by ELISA, as shown in Fig 1. Compared with the DMSO group, the expression of IL-1β and IL-6 was significantly elevated in the LPS group and LPS+RGZ group. RGZ significantly reduced the expression of IL-1β and IL-6 in LPS-treated animals.

### The effect of RGZ on uterus tissue pathologic changes during LPS-induced endometritis

The effect of RGZ on LPS-induced histopathologic changes in uterus tissues of mice were shown in Fig 2. Both DMSO control and RGZ control groups exhibited a normal structures. In the LPS group, the uterus tissue showed markedly damage, including hyperemia, hemorrhage, infiltration of many neutrophils in the uterus, and shedding of epithelial cells. Compared with the LPS group, the degree of tissue loss in the RGZ+LPS group was reduced.

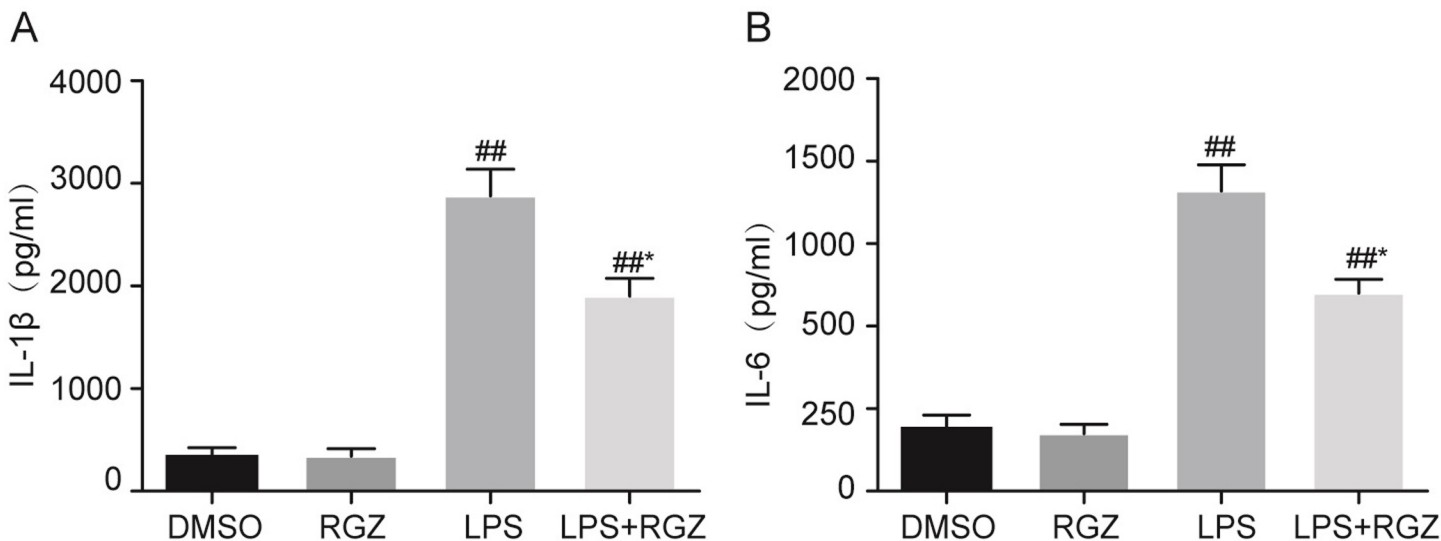

**Fig 1. Effects of RGZ on the expression of inflammatory cytokine in the uterine tissue.** Compared with the DMSO group, the expression of IL-1β and IL-6 was significantly elevated in the LPS group and LPS+RGZ group. RGZ significantly reduced the expression of IL-1β and IL-6 in LPS-treated animals. The values presented are the mean ± SEM (n = 10 in each group). $^{##}P < 0.01$ vs. DMSO group. $^{*}P < 0.05$, vs. LPS group.

Table 2 was the histopathologic scoring criteria [19]. A cumulative score combining hyperemia and Polymorphonuclear cell (PMN) infiltration represented the total points.

Student's *t*-test was used to statistically analyze the histological scores of the control groups and the experimental test groups. The results showed that there was no difference

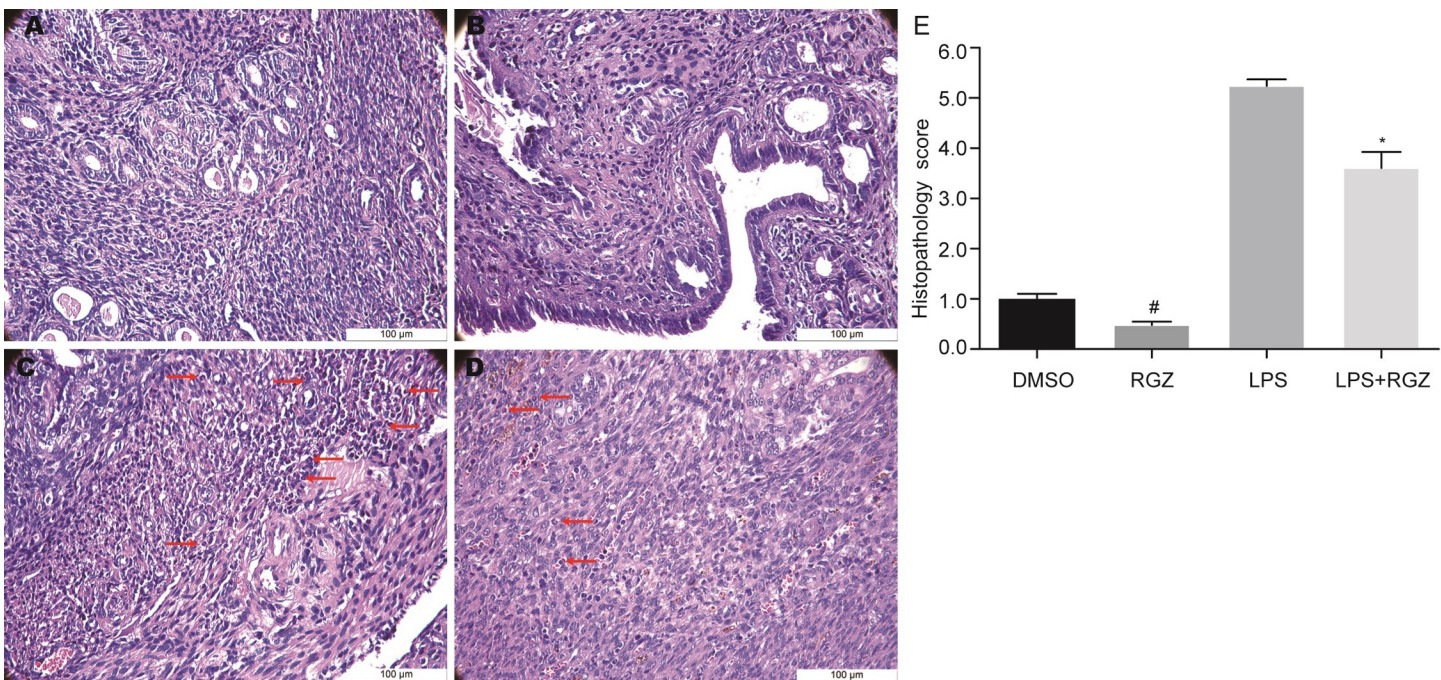

**Fig 2. The effect of RGZ on uterus tissue pathologic changes during LPS-induced endometritis for H&E staining.** Scale bar: 100 μm. Mice were given an intraperitoneal injection of RGZ (10 mg/kg) 2 hours prior to uterine perfusion of LPS. Uterus tissue was collected 24 h following LPS challenge. **a** DMSO control group, **b** RGZ control group, **c** LPS group, **d** LPS + RGZ group. Neutrophils were indicated by the arrows. (**e**) Inflammation score of uterus tissue. The histopathologic scoring was a cumulative score of hyperemia and neutrophils infiltration. The values presented were the means ± SEM (n = 10). ns, not significant, $^{#}P < 0.05$ and $^{*}P < 0.05$ compared with DMSO control group and LPS group, respectively.

**Table 2. The histopathologic scoring criteria.**

| Feature | Description | Score |
|---|---|---|
| Hyperemia/edema | Normal | 0 |
| | Mild | 1 |
| | Moderate | 2 |
| | Severe | 3 |
| Infiltration with neutrophil | 0–1 | 0 |
| | 2–5 | 1 |
| | 6–10 | 2 |
| | 11–15 | 3 |
| | 16–20 | 4 |
| | >20 | 5 |

between the control groups, and there was a significant difference between the LPS and LPS +RGZ groups, suggesting that RGZ could significantly improve the pathological changes of endometritis caused by LPS. To further explore the protective effect of RGZ on LPS-induced endometritis, we determined the expression of TLR4 in uterine tissue in different test groups. The immunohistochemical staining (IHC) results showed that the expression of TLR4 in the control groups was very low, with no difference between them. Additionally, the expression of TLR4 after RGZ treatment was significantly lower than that of the LPS group (Fig 3). Taken together, these results suggest that RGZ may suppress inflammation by inhibiting TLR4.

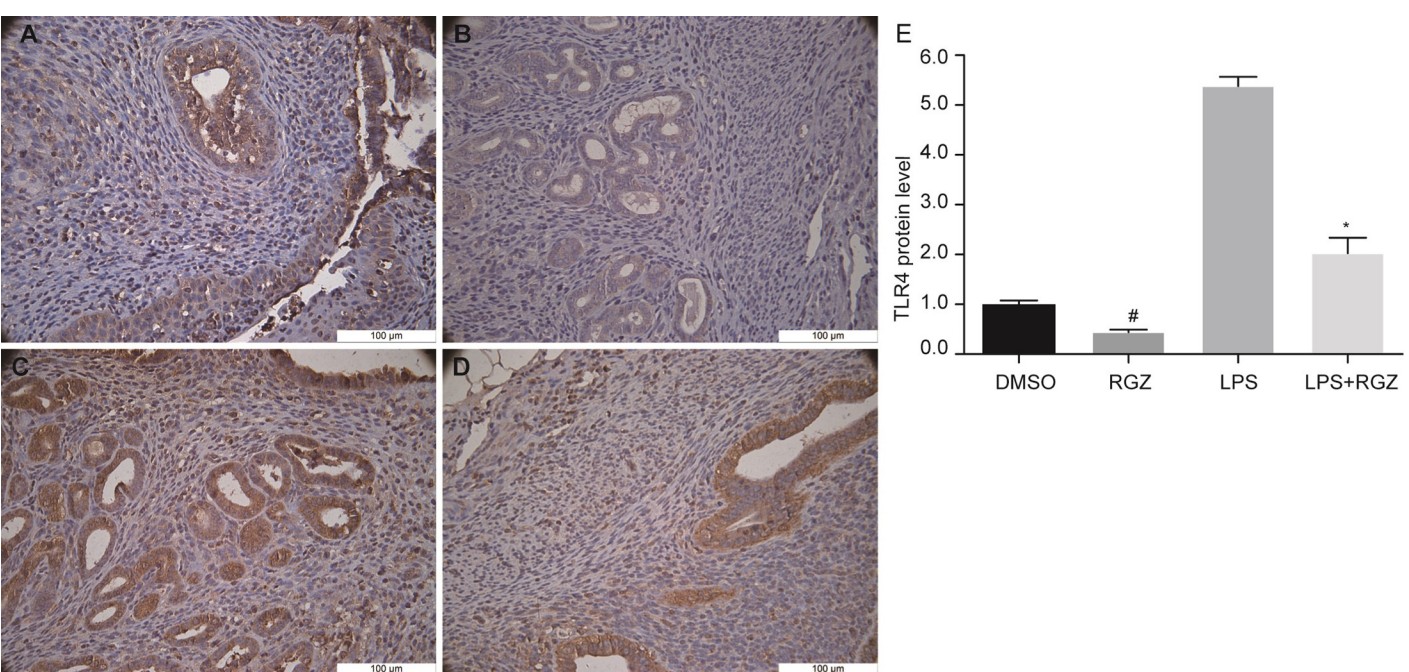

**Fig 3. Immunohistochemical assessment of TLR4 expression in endometrial tissue of mouse. a** DMSO control group, **b** RGZ control group, **c** LPS group, **d** LPS + RGZ group. TLR4 staining levels in endometrial tissues showed a reduced intensity in LPS + RGZ group compared to the LPS group. TLR4 staining levels in endometrial tissue indicated a reduced intensity in the RGZ group compared to the DMSO control group. The values presented were the means ± SEM (n = 10). ns, not significant, [#]$P < 0.05$ and [*]$P < 0.05$ compared with DMSO control group and LPS group, respectively.

### The effect of RGZ on TLR4 expression and phosphorylated IκBα and p65 during LPS-induced endometritis

As shown in Fig 4A and 4B, the LPS-stimulated group had the highest expression of TLR4 in uterine tissue compared to the other groups. This suggests that inflammation was most severe in this group. After treatment with RGZ, the expression of TLR4 in uterine tissue decreased significantly, indicating a statistically significant difference between the two groups. TLR4, as a receptor specifically recognized by gram-negative bacteria LPS, was overexpressed in LPS groups. The binding of LPS and TLR4 can induce intracellular signal transduction response, which activate the NF-κB signaling pathway and stimulate the expression of NF-κB signaling pathway related inflammatory factors. The expression of p-p65 and p-IκB in RGZ group was lower than that in LPS group due to its anti-inflammatory effect (Fig 4C and 4D). These results strongly suggest that RGZ inhibits the LPS-TLR4-pathway to reduce LPS-induced inflammation.

### Effects of RGZ on cell viability

Before conducting cell culture in vitro, it is necessary to first determine whether the drug has toxic effects on the cells to ensure the accuracy of the reaction caused by the tested drug. In this study, the cells were grown in experimental media containing various concentrations and the potential cytotoxic effects of RGZ on HESCs were examined using the CCK-8 assay. The results are provided in Fig 5. After 24 hours, there was no significant difference in the optical density values between the groups of cells after exposure to RGZ treatment below 20 μM/L. The results of the CCK-8 assay suggested that RGZ at the concentrations used (10 μM/L and 20 μM/L) had no cytotoxic effect on HESCs ($p = 0.15$ and $p = 0.15$), but RGZ at the concentrations of 40 μM/L had a significant cytotoxic effect on HESCs ($p = 0.04$).

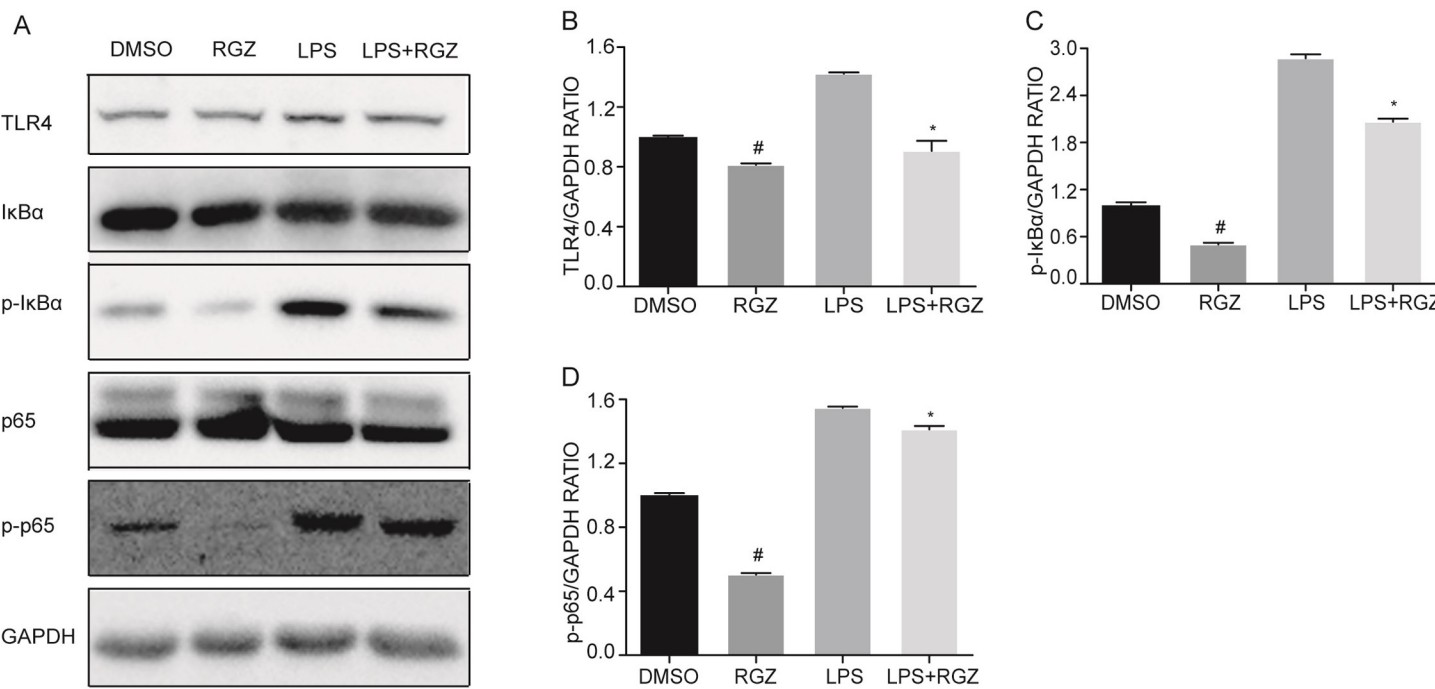

**Fig 4. The effects of RGZ on LPS-induced activation of TLR4 and NF-κB in vivo.** The protein levels of TLR4, p-p65 and p-IκBα in uterus tissues were measured by Western blotting. Data were represented the mean ± SEM (n = 10). #$P < 0.05$ and *$P < 0.05$ compared with DMSO control group and LPS group, respectively.

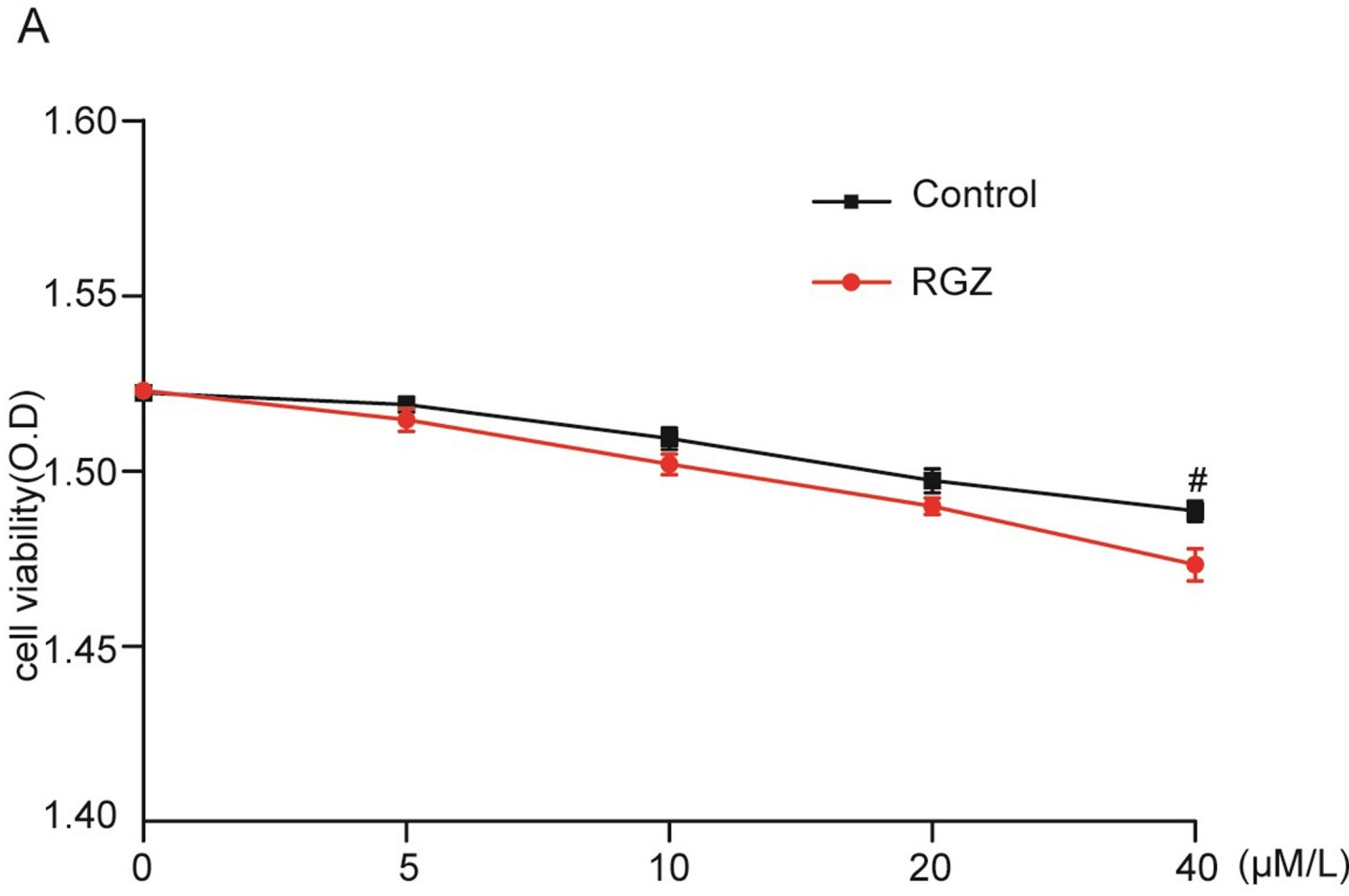

**Fig 5. The effects of RGZ on the cell viability of HESCs.** Cells were treated with the indicated concentration of RGZ (0~40 μM/L) for 24 h, and the cell viability was assessed by CCK-8 kits. $^{\#}P < 0.05$ compared with control group.

### Effects of RGZ on cell cytokines production

To explore the potential anti-inflammatory effect of RGZ in vitro. We initially treated the cells with different concentrations of LPS (0.01, 0.05, 0.25, 1,5 μg/mL) for 24 h to mimic the inflammatory state induced by LPS in vitro. Since both IL-1β and IL-6 expressed the highest at LPS concentration of 1μg/mL, the subsequent experiments used LPS at a concentration of 1μg/mL (Fig 6A and 6B). Next, we measured the relative genes expression of IL-1β and IL-6, and the results showed that the levels of IL-1β, IL-6 and TLR4 were decreased in a dose-dependent manner by RGZ treatment (Fig 6C–6E). The TLR4 results were consistent with the immunohistochemical staining and Western blot results of the LPS induced group.

### RGZ blunts LPS-induced activation of TLR4 and NF-κB in vitro

As displayed in Fig 7 the levels of p-p65 and Phospho-IκBα (p-IκBα) were noticeably increased in the LPS group compared to the control group. The levels of TLR4, p-p65 and Phospho-IκBα (p-IκBα) were noticeably increased in the LPS group compared to the control group. Their levels were greatly reduced in the RGZ-treated group and RGZ+TAK-242 treated group. Moreover, immunofluorescence staining showed the nuclear translocation of

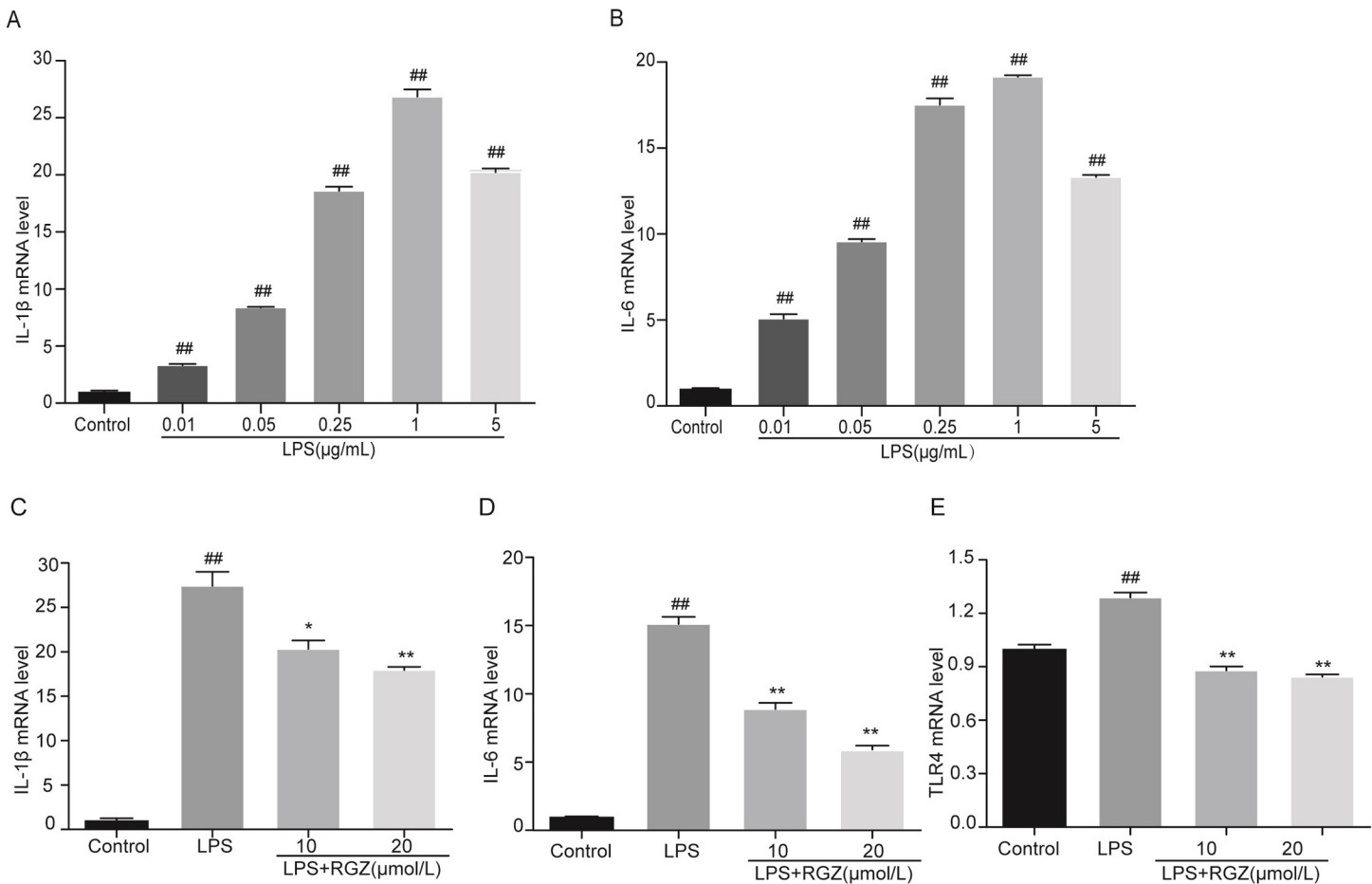

**Fig 6. mRNA levels of cytokines and TLR4 expression from HESCs stimulated by LPS and RGZ. a-b** mRNA levels of IL-1β and IL-6 in LPS-stimulated HESCs. GAPDH was used as a control. Control is Effects of RGZ on cytokines and TLR4 expression. the control group; LPS (0.01,0.05,0.25,1,5μg/mL) are the LPS groups. **c-e** Control is the control group; LPS (1μg/mL) is the LPS group; and 10 and 20 are the RGZ treatment groups representing 10 μM/L and 20 μM/L per cell plate, respectively. The data presented were the means ± SEM. $^{##}P < 0.01$, $^{*}P < 0.05$ and $^{**}P < 0.01$ compared with control group and LPS group, respectively.

NF-κB p65 with different treatments (Fig 8). These observations further confirmed the inhibitory effect of RGZ on NF-κB activation, which is in full agreement with the in vivo findings.

## Discussion

In recent years, animal models of LPS-induced endometritis have been widely used in the study of this disease [18, 20]. In this study, intrauterine infusion of LPS was used in mice to construct an endometritis model. We found that after LPS exposure, the mouse endometrium showed pathological changes and obvious inflammatory cell infiltration, indicating the successfully establishment of the endometritis mouse model through intrauterine LPS injection. Intraperitoneal injection of RGZ before LPS perfusion significantly relieved endometritis, along with decreased inflammatory cell infiltration and decreased TLR4 expression. These results indicated that RGZ inhibited LPS-induced endometritis.

In this study, the toxic effect of RGZ on HESCs was evaluated by MTT assay. RGZ treatment did not significantly affect the normal growth activity of HESC until the concentration

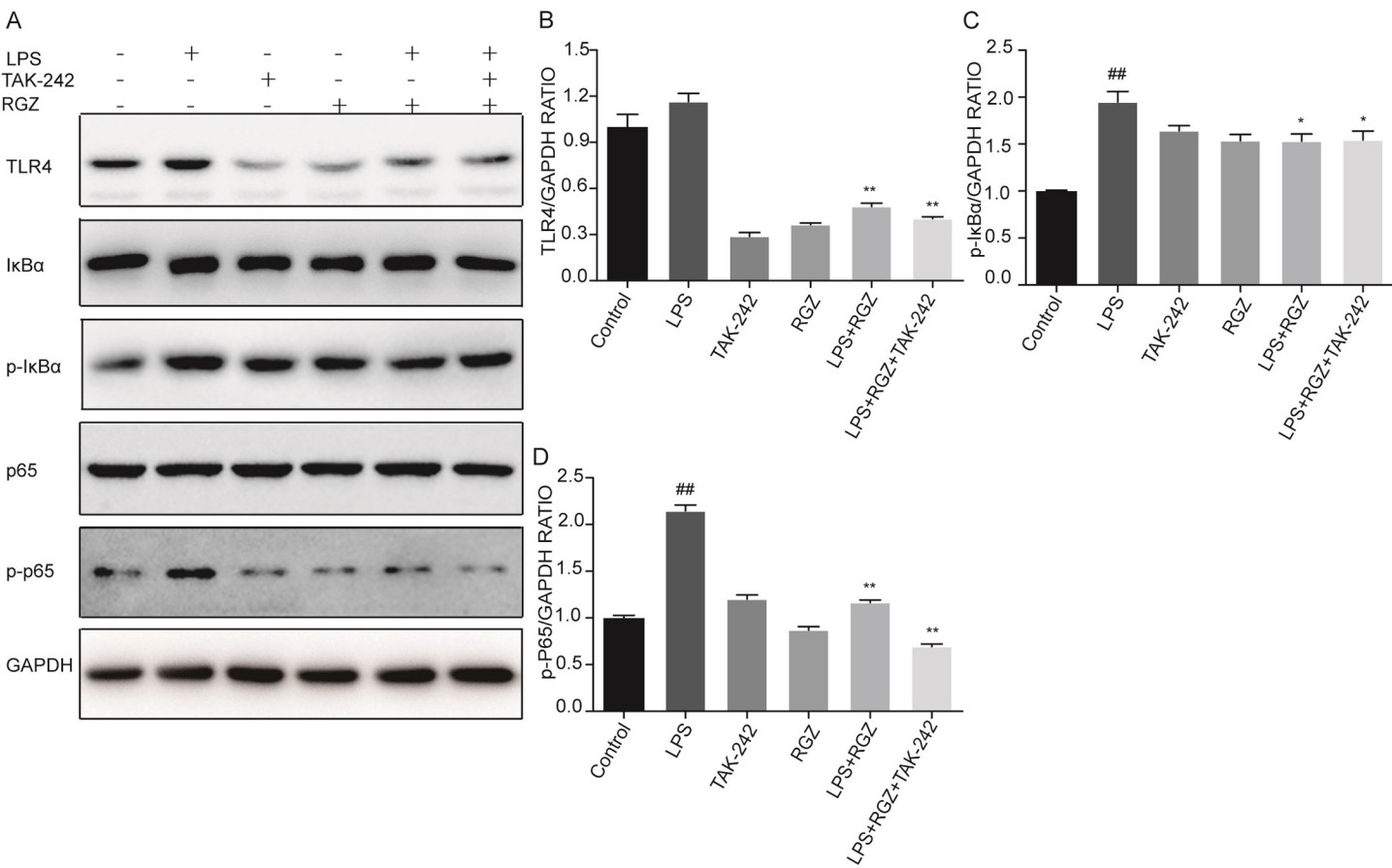

**Fig 7. The effects of RGZ and TAK-242 on LPS-induced activation of TLR4 and NF-κB in vitro.** The protein levels of TLR4, p-p65 and p-IκBα in HESCs were measured by Western blotting. GAPDH served as the control. Control is the control group; LPS is the LPS group; TAK-242 was the 1 μM/L TAK-242 treatment groups per cell plate, RGZ was the 10 μM/L RGZ treatment groups per cell plate, LPS+RGZ was the 30min incubation of the cell pretreatment with 10 μM RGZ followed by LPS induced for 24 h incubation, LPS+RGZ+TAK-242 was the 2 h incubation of 1 μM TAK-242 combined with 30min incubation of with 10 μM RGZ followed by LPS induced for 24 h incubation. Data represent the mean ± SEM. $^{##}P < 0.01$, $^{*}P < 0.05$ and $^{**}P < 0.01$ compared with control group and LPS group, respectively.

of RGZ reached 40 μM/L, at which point cell viability decreased significantly. This demonstrates that the drug concentrations selected for treatment in this study did not cause significant cytotoxicity. Under normal conditions, p65 binds to IκBα in the cytoplasm. Upon stimulation by LPS, IκBα is phosphorylated and degraded, while p65 is phosphorylated and enters the nucleus to promote the transcription of nuclear genes, leading to the release of a large number of inflammatory factors [21, 22]. In this study we found that LPS dose-dependently induced HESC to secrete IL-1β and IL-6, while RGZ dose-dependently reduced LPS-induced IκBα and p65 phosphorylation, inhibited the activation of NF-κB signaling pathway, and inhibited the expression of IL-1β and IL-6.

As a key receptor of LPS, TLR4 mediates immune response, which can initiate a cascade of downstream factors, regulate the synthesis and expression of downstream inflammatory mediators, and further activates the translocation of nuclear factor NF-κB to trigger an inflammatory response. Therefore, inhibiting the activation of TLR4/NF-κB signaling pathway proteins may be a target for the treatment of inflammation [23, 24]. We found in this study that RGZ not only reduced the expression of TLR4 in mouse endometrium but also down-regulated TLR4 in HESC cells, accompanied by a decrease in the expression of inflammatory factors, which were similar to the effects of a TLR4 inhibitor (Fig 6). Jiang

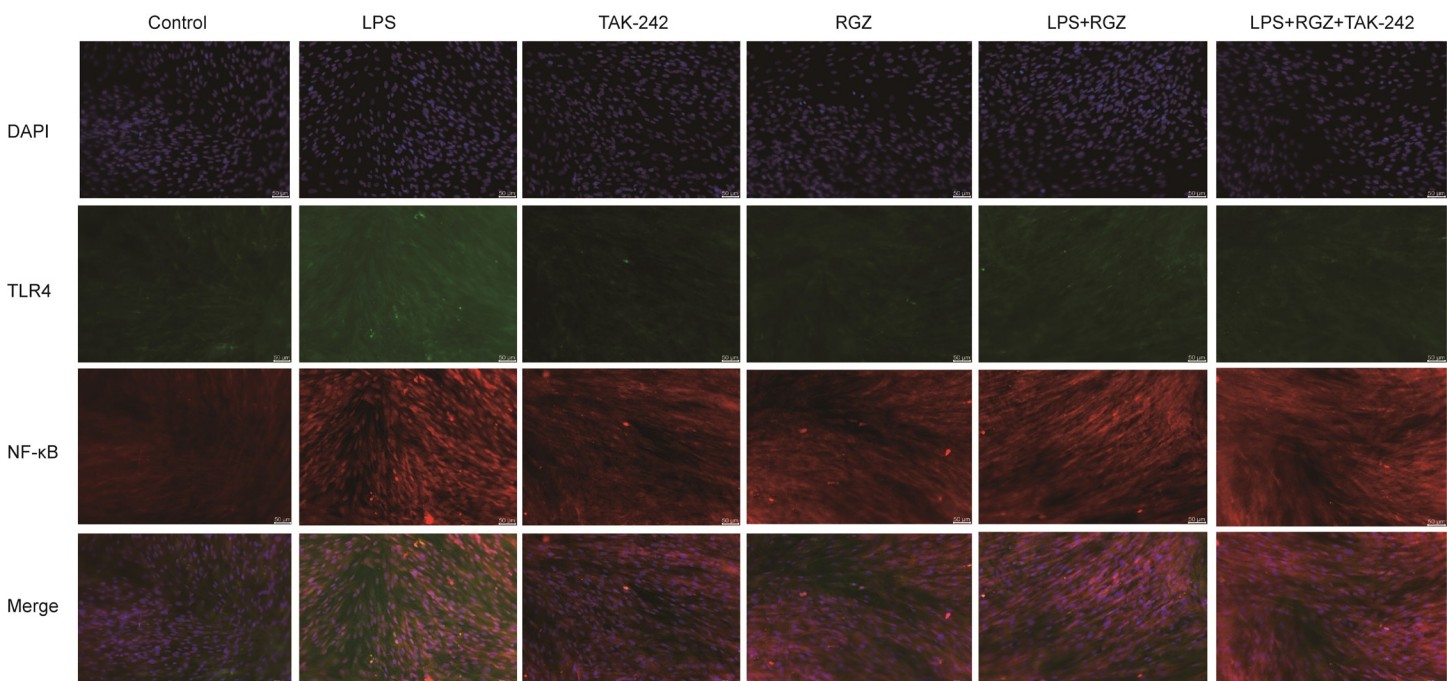

**Fig 8. RGZ inhibit NF-κB p65 translocation into the nucleus and expression of TLR4 in vitro.** Translocation of the NF-κB p65 subunit from the cytoplasm into the nucleus was assessed by immunofluorescence staining. After LPS stimulation, TLR4 was markedly increased in uterus tissue compared to the control group. Treatment with RGZ and RGZ+TAK-242 reduced the expression of TLR4. Scale bar: 50 μm. Blue spots represent cell nuclei, and green spots indicate TLR4 staining. red spots indicate p-p65 staining.

et al. reported that TLR4 knockout significantly reduced the phosphorylation level of p65 and the expression of IL-1β and TNF-α after LPS stimulation [25]. Kadam et al. reported that in an LPS-induced preterm birth model, RGZ reversed the inflammatory effect of LPS to prevent preterm birth by down-regulating TLR4 and up-regulating antioxidant responses [18]. RGZ inhibits LPS-induced inflammation by reducing pro-inflammatory factor production through the TLR4/NF-κB signaling pathway [26, 27]. Although RGZ is an agonist of PPARγ, and the activation of PPARγ is known to have anti-inflammatory effects [28, 29], PPARγ was not detected in HESCs by Western blot in this study due to the limitation of experimental conditions and the low expression of PPARγ in HESCs. Therefore, it remains unclear whether RGZ exerts anti-inflammatory effects through PPARγ receptors in addition to the TLR4/NF-κB pathway in HESCs.

In conclusion, LPS can induce the secretion of inflammatory factors by HESCs, thereby contributing to the initiation and development of endometritis. RGZ not only inhibited endometritis in mice but also dose-dependent inhibited the production of inflammatory factors induced by LPS in HESCs. These results indicated that RGZ exerted its potential protective effects on LPS-stimulated endometritis by inhibiting of TLR4/NF-κB-mediated inflammatory pathway (Fig 9). These findings suggest that RGZ has the potential to treat endometritis. RGZ may become a new drug for the treatment of endometritis, and the in-depth study of its mechanism of action has important theoretical significance for the prevention and treatment of this disease.

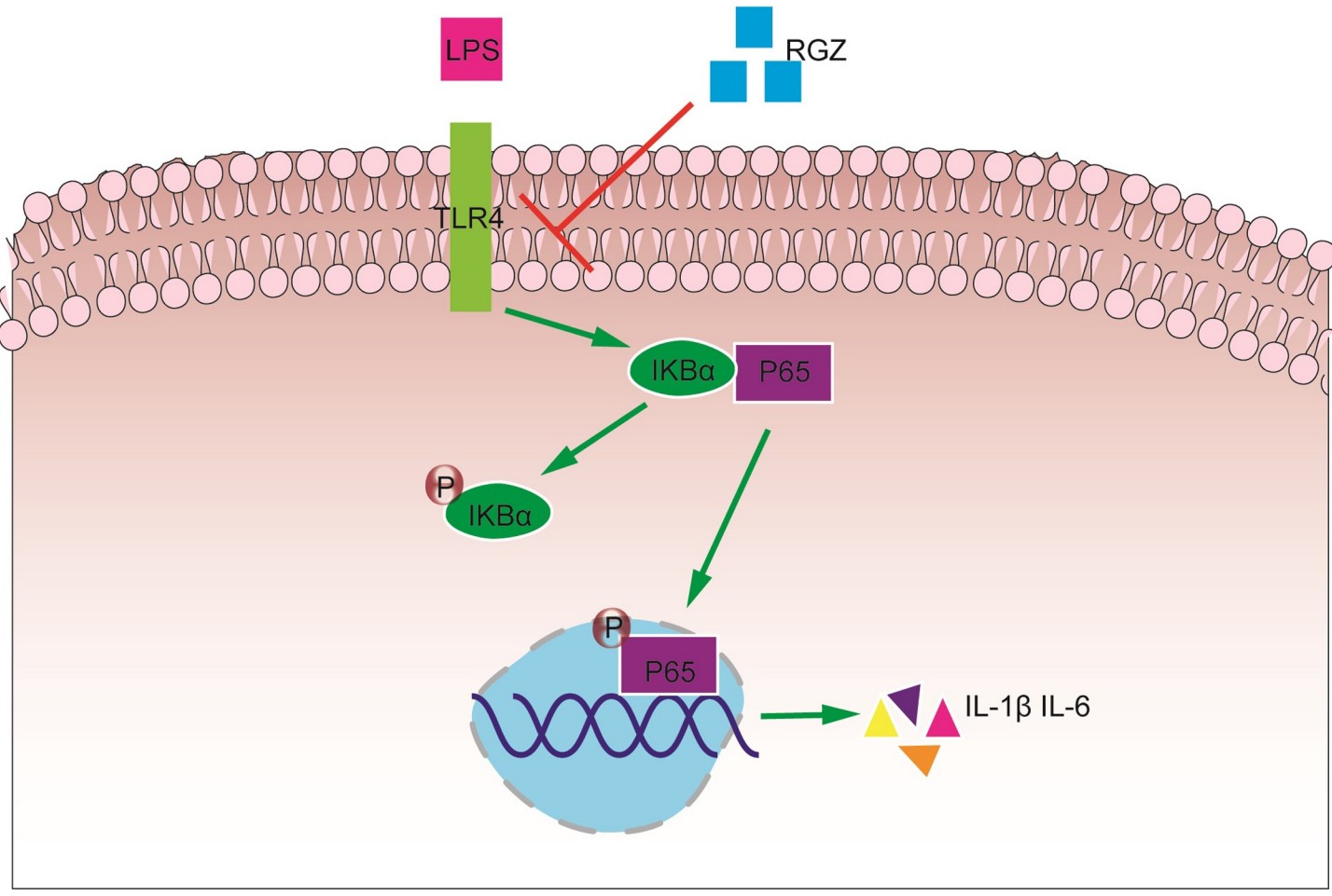

**Fig 9. Schematic diagram of signaling pathways related to the anti-inflammatory effects of RGZ on LPS-induced inflammation.**

## Supporting information

**S1 Raw images.**
(PDF)

## Author Contributions

**Conceptualization:** Ding Ma.

**Data curation:** Hongchu Bao, Jianxiang Cong, Qinglan Qu.

**Formal analysis:** Jianxiang Cong.

**Funding acquisition:** Ding Ma.

**Investigation:** Hongchu Bao, Qinglan Qu, Shunzhi He, Dongmei Zhao.

**Methodology:** Shunzhi He, Dongmei Zhao.

**Supervision:** Ding Ma.

**Validation:** Shuyuan Yin.

**Visualization:** Huishan Zhao.

**Writing – original draft:** Huishan Zhao, Shuyuan Yin.

**Writing – review & editing:** Ding Ma.

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
