## [Decision Letter · Decision Letter 0]

14 Jun 2023

PONE-D-22-35407Rosiglitazone alleviates LPS-induced endometritis via suppression of TLR4-mediated NF-κB activationPLOS ONE

Dear Dr. Ma,

Thank you for submitting your manuscript to PLOS ONE. After careful consideration, we feel that it has merit but does not fully meet PLOS ONE’s publication criteria as it currently stands. Therefore, we invite you to submit a revised version of the manuscript that addresses the points raised during the review process.

We look forward to receiving your revised manuscript.

Kind regards,

Lingshan Gou, Ph.D.

Academic Editor

PLOS ONE

Journal Requirements:

3. To comply with PLOS ONE submissions requirements, in your Methods section, please provide additional information regarding the experiments involving animals and ensure you have included details on (1) methods of sacrifice, (2) methods of anesthesia and/or analgesia, and (3) efforts to alleviate suffering.

4. As part of your revision, please complete and submit a copy of the Full ARRIVE 2.0 Guidelines checklist, a document that aims to improve experimental reporting and reproducibility of animal studies for purposes of post-publication data analysis and reproducibility: https://arriveguidelines.org/sites/arrive/files/documents/Author%20Checklist%20-%20Full.pdf (PDF). Please include your completed checklist as a Supporting Information file. Note that if your paper is accepted for publication, this checklist will be published as part of your article.

5. Thank you for providing an English translation for the ethics approval document titled "Animal Ethics declarations-2" with the approval number "2022 -255". In your ethics statement in the online submission form and the Methods section, please refer to the above mentioned document "2022 -255" and state that your study received ethics approval according to this document. For this purpose, please also mention the approval number "2022 -255".

6. Thank you for stating the following financial disclosure: 

The study was supported by Shandong Medical and Health Science and Technology Development Project (No.202005030944).

The study was supported by Shandong Medical and Health Science and Technology Development Project (No.202005030944).

7. Your ethics statement should only appear in the Methods section of your manuscript. If your ethics statement is written in any section besides the Methods, please delete it from any other section. 

Additional Editor Comments:

Comments and suggestions of Reviewer 1:

Manuscript Number: PONE-D-22-35407

Ding Ma et al.,

Rosiglitazone alleviates LPS-induced endometritis via suppression of TLR4-mediated

NF-κB activation

The study is well designed and organized. The reviewer suggests that the manuscript be checked for the grammatical and typographical errors.

Comments and suggestions of Reviewer 2:

The authors’ manuscript is interesting, but there are still some minor problems:

1. In the histopathologic results, the authors mentioned the histological scores, perhaps they should explain how they were calculated.

2. In WB results, the authors mentioned that “overexpression of TLR4 weakened the effect of RGZ on LPS-induced p65 phosphorylation and IκB expression in HESCs” which is not very understandable here.

3. The results for cell viability and cytokines production should be divided into two sections.

4. In cytokines production results, the authors mention the protein expression of IL-1β and IL-6, where it is better to write the relative expression of genes. In addition, the authors said that consistent with the in vivo results, there is no in vivo results were seen.

5. Fig.5 legend, mRNA should not be capitalized.

Editor's comments:

1. The written should be improved and the grammar errors needed to be corrected.

2. In Table 1, the primer is in human or in mouse, it needed to be clearly indicated in Table 1.

3. How do you design the dose of RGZ in in vivo studies?

4. TLR4 affected by RGZ demonstrated in this study. Whether RGZ affected other TLR family numbers, such TLR1, TLR2, etc？

Reviewers' comments:

Reviewer's Responses to Questions

**Comments to the Author**

1. Is the manuscript technically sound, and do the data support the conclusions?

Reviewer #1: Yes

Reviewer #2: Yes

2. Has the statistical analysis been performed appropriately and rigorously? 

Reviewer #1: Yes

Reviewer #2: N/A

3. Have the authors made all data underlying the findings in their manuscript fully available?

Reviewer #1: Yes

Reviewer #2: Yes

4. Is the manuscript presented in an intelligible fashion and written in standard English?

Reviewer #1: Yes

Reviewer #2: Yes

5. Review Comments to the Author

Reviewer #1: Manuscript Number: PONE-D-22-35407

Ding Ma et al.,

Rosiglitazone alleviates LPS-induced endometritis via suppression of TLR4-mediated

NF-κB activation

The study is well designed and organized. The reviewer suggests that the manuscript be checked for the grammatical and typographical errors.

Reviewer #2: The authors’ manuscript is interesting, but there are still some minor problems:

1. In the histopathologic results, the authors mentioned the histological scores, perhaps they should explain how they were calculated.

2. In WB results, the authors mentioned that “overexpression of TLR4 weakened the effect of RGZ on LPS-induced p65 phosphorylation and IκB expression in HESCs” which is not very understandable here.

3. The results for cell viability and cytokines production should be divided into two sections.

4. In cytokines production results, the authors mention the protein expression of IL-1β and IL-6, where it is better to write the relative expression of genes. In addition, the authors said that consistent with the in vivo results, there is no in vivo results were seen.

5. Fig.5 legend, mRNA should not be capitalized.

6. PLOS authors have the option to publish the peer review history of their article (what does this mean?). If published, this will include your full peer review and any attached files.

Reviewer #1: No

Reviewer #2: No

---

## [Author Response · Author response to Decision Letter 0]

20 Jul 2023

Dear Editors and Reviewers:

We greatly appreciate your and other reviewers' criticism and thoughtful suggestions. These valuable opinions will help improve the quality of the manuscript. We have carefully considered these opinions and made corresponding modifications, hoping for approval. The revised manuscript has been adjusted to the requirements of the journal. All changes made to the manuscript are highlighted in red for easy recognition. The following are the main corrections and responses to the reviewer's comments:

Response to comments of Editor：

1. The written should be improved and the grammar errors needed to be corrected.

Response: Thank you very much for your sincere suggestions. According to your suggestions, we have carefully revised the language format of the article and invited relevant professionals to review and correct it.

2. In Table 1, the primer is in human or in mouse, it needed to be clearly indicated in Table 1.

Response: Thank you very much for your careful review of this manuscript. In order for other readers to have a clearer understanding of the procedures of this study, we have clearly stated the source of primers in the manuscript according to your suggestion.

3. How do you design the dose of RGZ in in vivo studies?

Response: Thank you very much for your comments. As mentioned in the fifth line of “Animals and treatment” in the manuscript, we set the experimental dosage of this project with reference to previous relevant research reports.

4. TLR4 affected by RGZ demonstrated in this study. Whether RGZ affected other TLR family numbers, such TLR1, TLR2, etc？

Response: Thank you again for your careful reading and professional questions. Many studies have shown that Toll-like receptors (TLRs) are part of the innate immune system and participate in the response to microbial pathogens. It has been reported that normal skin epidermal keratinocytes compositively express TLR1, TLR2 recognizes a variety of ligands expressed by Gram-positive bacteria, and TLR3, TLR4 and TLR5 specifically recognize double-stranded RNA, Gram-negative lipopolaccharide and bacterial flagellin, respectively. In this study, endometritis was mainly caused by Gram-negative bacteria, so the TLR4 receptor was mainly studied. As for the other receptors you mentioned, this gives us an idea for other things we can do with RGZ in the future.

Response to comments of Reviewer 1：

The study is well designed and organized. The reviewer suggests that the manuscript be checked for the grammatical and typographical errors.

Response: Thank you very much for your valuable recommendations. We apologize for our grammatical errors and typographical errors. In accordance with the requirements of the journal, we carefully checked the grammar in the revised manuscript and asked professionals to make corrections. 

Response to comments of Reviewer 2：

1. In the histopathologic results, the authors mentioned the histological scores, perhaps they should explain how they were calculated.

Response: Thank you very much for your suggestion. We are sorry for the trouble caused by our simple expression. Based on your professional advice, we have provided a detailed description of the grading criteria for histopathology score in the revised manuscript.

2. In WB results, the authors mentioned that “overexpression of TLR4 weakened the effect of RGZ on LPS-induced p65 phosphorylation and IκB expression in HESCs” which is not very understandable here.

Response: Thank you very much for your suggestion. We apologize for the confusion caused by our improper expression. We have rephrased the WB results in the revised manuscript to facilitate readers' clearer understanding of this mechanism.

3. The results for cell viability and cytokines production should be divided into two sections. 

Response: Thank you very much for your valuable recommendations. We accepted your suggestion and divided cell viability and cytokines production into two parts in the revised manuscript.

4. In cytokines production results, the authors mention the protein expression of IL-1β and IL-6, where it is better to write the relative expression of genes. In addition, the authors said that consistent with the in vivo results, there is no in vivo results were seen.

Response: Thank you very much for your professional comments. 

We accept your professional suggestions and have made modifications in the revised manuscript according to your suggestions. At the same time, we are sorry for your misunderstanding caused by our inappropriate description. In the revised manuscript, we restated this result to prevent confusion among other readers. We collected serum from mice and detected the expression of IL-1β and IL-6 using ELISA method β, but there was no statistically significant difference between the groups, possibly due to the lack of systemic response caused by local inflammation of the endometrium. This is consistent with the conclusion reported in the literature that acute phase protein assay has not proved useful in the diagnosis of endometritis[1]. Therefore, the results were not included in the manuscript. 

[1] Sikora M , Król, Jaros aw, Nowak M ,et al.The usefulness of uterine lavage and acute phase protein levels as a diagnostic tool for subclinical endometritis in Icelandic mares.Acta Veterinaria Scandinavica, 2015, 58(1):50. DOI:10.1186/s13028-016-0233-4.

5. Fig.5 legend, mRNA should not be capitalized.

Response: Thank you very much for your advice. We have corrected this error in the revised manuscript.

---

## [Decision Letter · Decision Letter 1]

29 Aug 2023

PONE-D-22-35407R1Rosiglitazone alleviates LPS-induced endometritis via suppression of TLR4-mediated NF-κB activationPLOS ONE

Dear Dr. Ma,

Thank you for submitting your manuscript to PLOS ONE. After careful consideration, we feel that it has merit but does not fully meet PLOS ONE’s publication criteria as it currently stands. Therefore, we invite you to submit a revised version of the manuscript that addresses the points raised during the review process.

We look forward to receiving your revised manuscript.

Kind regards,

Lingshan Gou, Ph.D.

Academic Editor

PLOS ONE

Reviewers' comments:

Reviewer's Responses to Questions

**Comments to the Author**

1. If the authors have adequately addressed your comments raised in a previous round of review and you feel that this manuscript is now acceptable for publication, you may indicate that here to bypass the “Comments to the Author” section, enter your conflict of interest statement in the “Confidential to Editor” section, and submit your "Accept" recommendation.

Reviewer #2: All comments have been addressed

Reviewer #3: (No Response)

2. Is the manuscript technically sound, and do the data support the conclusions?

Reviewer #2: Yes

Reviewer #3: Partly

3. Has the statistical analysis been performed appropriately and rigorously? 

Reviewer #2: Yes

Reviewer #3: Yes

4. Have the authors made all data underlying the findings in their manuscript fully available?

Reviewer #2: Yes

Reviewer #3: Yes

5. Is the manuscript presented in an intelligible fashion and written in standard English?

Reviewer #2: Yes

Reviewer #3: Yes

6. Review Comments to the Author

Reviewer #2: (No Response)

Reviewer #3: 1. In order to further confirm the effect of rosiglitazone on the Nf-kb pathway, it is recommended to add immunofluorescence staining to observe the nuclear translocation of Nf-kb.

2. It is recommended to add TLR4 inhibitors to observe whether they have the same protective effect.

3. It is recommended to detect the level of inflammatory factors in animal models, rather than judging the success of modeling by morphological changes of tissues.

7. PLOS authors have the option to publish the peer review history of their article (what does this mean?). If published, this will include your full peer review and any attached files.

Reviewer #2: No

Reviewer #3: No

---

## [Author Response · Author response to Decision Letter 1]

17 Oct 2023

Dear Editors and Reviewers:

We are delighted to receive your comments and suggestions from you and the reviewers. We have carefully considered these comments and made corrections accordingly, which we hope will be approved for publication. All changes made to the text are highlighted in green for easy identification. The main revisions and replies to the reviewer's comments are as follows:

Response to comments of Reviewer 3：

1. In order to further confirm the effect of rosiglitazone on the Nf-kb pathway, it is recommended to add immunofluorescence staining to observe the nuclear translocation of NF-κB.

Response: Thank you very much for your valuable suggestion. Based on your feedback, we have supplemented the results of immunofluorescence staining to observe the nuclear translocation of NF-κB and described them in the manuscript. The results are shown in Figure 7.

2. It is recommended to add TLR4 inhibitors to observe whether they have the same protective effect.

Response: Thank you for your valuable advice. We accept your proposal. We added this part of the experiment in the revised manuscript to ensure the reliability of the experimental results.

3. It is recommended to detect the level of inflammatory factors in animal models, rather than judging the success of modeling by morphological changes of tissues.

Response: Thank you very much for your valuable advice. We have supplemented this experiment in the experimental section of the manuscript to further ensure experimental rigor.

---

## [Decision Letter · Decision Letter 2]

21 Nov 2023

Rosiglitazone alleviates LPS-induced endometritis via suppression of TLR4-mediated NF-κB activation

PONE-D-22-35407R2

Dear Dr. Ma,

We’re pleased to inform you that your manuscript has been judged scientifically suitable for publication and will be formally accepted for publication once it meets all outstanding technical requirements.

Kind regards,

Lingshan Gou, Ph.D.

Academic Editor

PLOS ONE

Additional Editor Comments (optional):

Reviewers' comments:

Reviewer's Responses to Questions

**Comments to the Author**

1. If the authors have adequately addressed your comments raised in a previous round of review and you feel that this manuscript is now acceptable for publication, you may indicate that here to bypass the “Comments to the Author” section, enter your conflict of interest statement in the “Confidential to Editor” section, and submit your "Accept" recommendation.

Reviewer #3: All comments have been addressed

2. Is the manuscript technically sound, and do the data support the conclusions?

Reviewer #3: Yes

3. Has the statistical analysis been performed appropriately and rigorously? 

Reviewer #3: Yes

4. Have the authors made all data underlying the findings in their manuscript fully available?

Reviewer #3: Yes

5. Is the manuscript presented in an intelligible fashion and written in standard English?

Reviewer #3: Yes

6. Review Comments to the Author

Reviewer #3: Although this manuscript has a very new theme, it is still debatable whether it can be applied in clinical practice. Because rosiglitazone is a drug for the treatment of diabetes, it has a significant effect on reducing blood glucose. If it is used for the treatment of other diseases, the effect of reducing blood glucose will become a side effect of the drug. This safety is worth careful consideration.

7. PLOS authors have the option to publish the peer review history of their article (what does this mean?). If published, this will include your full peer review and any attached files.

Reviewer #3: **Yes: **Shuaishuai Wang
